# Segmentation and Phenotype Calculation of Rapeseed Pods Based on YOLO v8 and Mask R-Convolution Neural Networks

**DOI:** 10.3390/plants12183328

**Published:** 2023-09-20

**Authors:** Nan Wang, Hongbo Liu, Yicheng Li, Weijun Zhou, Mingquan Ding

**Affiliations:** 1The Key Laboratory for Quality Improvement of Agricultural Products of Zhejiang Province, College of Advanced Agricultural Sciences, Zhejiang A&F University, Linan, Hangzhou 311300, China; jerrywang1010@sina.com (N.W.); hbliu@zafu.edu.cn (H.L.);; 2Institute of Crop Science and Zhejiang Key Laboratory of Crop Germplasm, Zhejiang University, Hangzhou 310058, China; wjzhou@zju.edu.cn

**Keywords:** instance segmentation, deep learning, rapeseed pods, length and width measure, counting

## Abstract

Rapeseed is a significant oil crop, and the size and length of its pods affect its productivity. However, manually counting the number of rapeseed pods and measuring the length, width, and area of the pod takes time and effort, especially when there are hundreds of rapeseed resources to be assessed. This work created two state-of-the-art deep learning-based methods to identify rapeseed pods and related pod attributes, which are then implemented in rapeseed pots to improve the accuracy of the rapeseed yield estimate. One of these methods is YOLO v8, and the other is the two-stage model Mask R-CNN based on the framework Detectron2. The YOLO v8n model and the Mask R-CNN model with a Resnet101 backbone in Detectron2 both achieve precision rates exceeding 90%. The recognition results demonstrated that both models perform well when graphic images of rapeseed pods are segmented. In light of this, we developed a coin-based approach for estimating the size of rapeseed pods and tested it on a test dataset made up of nine different species of *Brassica napus* and one of *Brassica campestris* L. The correlation coefficients between manual measurement and machine vision measurement of length and width were calculated using statistical methods. The length regression coefficient of both methods was 0.991, and the width regression coefficient was 0.989. In conclusion, for the first time, we utilized deep learning techniques to identify the characteristics of rapeseed pods while concurrently establishing a dataset for rapeseed pods. Our suggested approaches were successful in segmenting and counting rapeseed pods precisely. Our approach offers breeders an effective strategy for digitally analyzing phenotypes and automating the identification and screening process, not only in rapeseed germplasm resources but also in leguminous plants, like soybeans that possess pods.

## 1. Introduction

*Brassica napus* (*B. napus*, AACC, 2n = 38), an annual heterotetraploidy plant that is native to Europe, including the Mediterranean basin and several agricultural areas of Northern and Western Europe since approximately 7500 years ago, was naturally crossed and doubled from two diploid species, namely, *Brassica oleracea* (AA, 2n = 18) and *Brassica rapa* (CC, 2n = 20) [1,2,3]. This plant is a key economic oilseed crop worldwide and accounts for approximately 13–16% of global vegetable oil production [4,5,6]. Currently, *Brassica napus* dominates rapeseed production in China and accounts for approximately 85% of the total rapeseed planting area [7,8]. Rapeseed oil is also a valuable feedstock for biofuel production, particularly biodiesel. The conversion of rapeseed oil into biodiesel offers a cleaner and more sustainable alternative to fossil fuels and contributes to the reduction in greenhouse gas emissions and air pollution. China, being one of the world’s largest energy consumers, has a vested interest in expanding its biofuel production capacity to mitigate environmental challenges and reduce its reliance on imported fossil fuels. However, with the growing demand for renewable energy sources and sustainable agricultural practices, the cultivation of rapeseed pods has gained renewed attention in the context of the energy industry in China [9,10].

The value of rapeseed seeds, which are a raw material in the oil industry, is strictly dependent on varietal yield parameters, which are one of the most important elements influencing crop production in China [11,12,13]. The rapeseed yield factor is becoming increasingly multifaceted as an important grain and oil crop [14]. Several factor components, such as plant type, plant height, the number of branches, inflorescence number, the number of fruits in sequence, the number of grains per fruit, and thousand kernel weight, can influence crop yield [15]. These components include pods, whose number, length, and width have a substantial influence on yield [16,17]. Thus, enhancement of production requires the selection of breeding materials with a high proportion of large siliques [18].

Traditional research on crop phenotypes relies on human observation experience, counting, weighing, and other manual measures, which typically last for a long period and require considerable human resources; in addition, the findings are often subjective and extremely inaccurate [19,20]. To solve this problem, scientists frequently employ computer vision and machine learning technology in crop breeding; these methods can be used for collection and analysis by high-throughput plant phenotypic imaging and offer a precise direction for breeding, variety selection, genomics, and phenomics. Moreover, as artificial intelligence (AI) algorithms have become popular in the last decade, computer vision and machine learning play a meaningful role in yield estimation, plant recognition and classification, and plant stress physiology [21,22]. Researchers used a deep learning (DL) algorithm to create a cell phone integration program for rapeseed pests and diseases [23], whereas Wen et al. used the random forest algorithm to create a yield estimation model for oilseed rape based on four years of localized observations in five different regions of eastern Canada and determined the optimal nitrogen application rate for various growing regions [22]. In addition, Du et al. presented the plant segmentation transformer network to segment dense point cloud data with complicated spatial structures gathered by handheld laser scanning in the pod stage of rapeseed [24]. Han et al. invented a deep learning net called InceptionV3-LSTM for the intelligent prediction of rapeseed harvest time, and a convolutional neural network model was applied for variety classification and seed quality assessment of winter rapeseed [25,26].

As a vital branch of machine learning, DL has been widely applied in image recognition and natural language processing [27,28]. DL, a kind of AI, has made advances in the study and use of plant phenotypes [29,30]. Convolution neural networks (CNNs) are feedforward neural networks that may successfully reduce the dimensions of a large amount of image information while preserving image characteristics [31,32]. With the progress of optoelectronic technology and mechanical technology, the combination of AI and planting has brought numerous methods to environmental sensing [33], such as unmanned aerial vehicles (UAVs) [34], hyperspectral imaging, computer vision, radar induction, and physicochemical analysis, to the study of plant phenotypic analysis. Combining UAVs, machine vision, and lidar technology, OSWSDet, a DL model, was applied to optimize traditional wheat ear image recognition [35]. The YOLO POD model was created to develop a production forecasting program to count soybean pods by multi-angle images on video data collected by field robots [36]. RiceNet, a model based on CNNs and high-throughput UAV RGB images, was built based on video data gathered by field robots for rice plant counting, location, and size evaluation [37].

Rapeseed yield is strongly connected to pod characteristics, such as the pod number, pod length, and pod width [38]; nevertheless, the present investigation approach is time-consuming and labor-expensive. At present, there are still the following problems in the cultivation of rapeseed pods, such as the lack of effective monitoring methods for the quantity and quality of rapeseed pods, and a lack of in-depth understanding of the growth process and influencing factors of rapeseed pods. Manually measured findings are frequently imprecise and contain substantial inaccuracies [39]. Given that rapeseed pods are generally narrow and have a particular bending angle, appropriate equipment for measurement is lacking, which results in inaccurate measurement findings.

In this paper, we present a simple and effective method to estimate the number, length, width, and two-dimensional image area of rapeseed pods using deep learning and computer vision and observe the length variation patterns of rapeseed pods in different time periods, such as the green ripening, yellow ripening, and mature periods. It was the first time that the most recent deep learning algorithms, YOLO v8 and Mask R-CNN, had been employed to detect yield correlation in rapeseed. Both models’ precision is above 90%; the length regression coefficient of both methods was 0.991 and the width regression coefficient was 0.989. This research will help us to deeply understand the natural growth pattern of rapeseed, improve its breeding and cultivation practices, and enhance its production and quality in the future. This study applies neural networks to analyze the phenotypic data information of rapeseed pods, which can be useful for industrial production. Meanwhile, the study methodologies described in this publication can provide a reference and guidance for research on other crops.

## 2. Results

### 2.1. Rapeseed Pod Detection Based on YOLO v8 Models

We collected 4461 rapeseed pods and randomly placed them on a black velvet background. We used a Canon EOS 800D camera to capture RGB images and enhance the data of the images, such as flipping, rotating, cropping, adding noise, blurring, masking, color transformation, cut out, and other methods, which aimed to increase the model’s generalization ability and avoid sample imbalance.

Three YOLOv8 configurations (n, s, and m) were first trained and tested to detect rapeseed pods. A 9:1 ratio was used to divide the dataset into training and validation sets. The model evaluation curve on precision confidence (P), recall confidence (R), mAP50 and mAP50-95 of the bound box and mask, FLOPs, parameters (M), and gradients of the model were obtained by training for 100 epochs (Table 1).

Table 1 demonstrates that as the model volume increased, along with the number of FLOPs, parameters, and gradients, precision, recall, and mAP50-95 progressively reached 1 in comparison with those of other models. When we applied the YOLOv8m model, the bound box’s precision confidence, the mask’s precision confidence, recall confidence, mAP50, and mAP50-95 peaked at 99.5%, 0.993, 0.998, 0.991, and 0.79, respectively. The segmentation result diagram is presented below, with the YOLOv8m model showing the best performance (Figure 1).

### 2.2. Rapeseed Pod Detection Based on the Mask R-CNN Model

In addition, we tested another cutting-edge model, the Mask R-CNN, which uses two stages of object detection. We utilized the identical dataset with YOLO v8 and performed tasks akin to those in Section 3.1 of our image processing work. The rapeseed pod instance segmentation project was built on top of the widely used and well-liked Detectron2 for easy comprehension and application. As a result, all the dataset registration training and testing were contained in a single script, which has the advantages of simple comprehension and suitability for the expansion of other models. Mask R-CNN can successfully identify rapeseed pods (Figure 2).

The training program included three components. Prior to direct registration, the Labelme annotation file was converted into a typical annotation JSON file for the COCO dataset. In addition, settings, such as learning rate and maximum iterations, were altered. The major modification in this case was to balance the data amount and avoid overfitting due to the lack of datasets available. We employed cosine annealing to tune the learning rate optimally to obtain the best results. The initial learning rate was set to 10^−3^, and Adam was selected as the optimizer after SGD and Adam were compared. By contrast, data augmentation techniques, such as flipping, mosaicking, and random scaling, were employed to improve the original data to overcome the issue of small data volume. Finally, modeling and detection were performed.

We selected the maximum iterations of 10,000 and 30,000 based on the Mask R-CNN model, whose backbones are ResNet50 and ResNet101, respectively. The difference between the two backbones is that ResNet101 has 51 more bottlenecks on conv4 than ResNet50. Table 1 shows that the bound box and segmentation’s average precision (AP), AP50 and AP75 values increased as the maximum iteration number increased. The AP50 and AP75 values indicate that the IoU thresholds for the average accuracy were greater than 0.5 and 0.75, respectively. The abbreviations APs, APm, and APl, which stand for “AP Small”, “AP Media”, and “AP Large”, respectively, denote the size of the object area. The higher the number of iterations, the higher the rate is regardless of the controlled variables. The metric rates also increased as the backbone was upgraded (Table 2).

### 2.3. Evaluation of the Proposed Rapeseed Pod Detection Model

We randomly selected samples of different *Brassica napus* and a limited number of *Brassica campestris* L. rapeseed pods. These pods are produced by numerous plants at different phases of growth. We employed a black velvet background and randomly selected test dataset photographs in various lighting circumstances, such as sunny or rainy days. The camera used was an iPhone 14 with a 3024 × 4032 or 4032 × 3024 image resolution and a 35 mm camera focal length.

We evaluated a test dataset of 38 images using the best model parameters and executed the test code using various YOLO v8 and Mask R-CNN parameter settings. The findings showed that Mask R-CNN outperformed YOLO v8 during actual testing, despite YOLO v8 having superior training model assessment metrics, including precision, recall, AP values, and others. When YOLOv8 was used, the rapeseed stem was confused with the rapeseed pod body. However, Mask R-CNN did not encounter this problem (Figure 3).

In addition, the Mask R-CNN model code developed based on the Detectron2 framework is more concise. Detectron2 has high flexibility and scalability, which allows for fast and direct training on single or multiple GPU servers while also helping researchers explore the most advanced algorithm designs effectively. Thus, Mask R-CNN has a faster testing speed than YOLO v8 (Table 3). A short testing period eases the application of algorithms to mobile devices and uses less computational power.

### 2.4. Detection of Rapeseed Pod Length, Width, and Cross-Sectional Area

We randomly selected different rapeseed pods and placed them on a black velvet cloth. Along with the rapeseed pods, we placed a one RMB coin with a 25 mm diameter made in 2012 by the People’s Bank of China. With the coin serving as a standard for real-world and image conversion, we used a collection of 18 images to measure the actual length, width, and area of two-dimensional images of rapeseed pods using machine vision. The steps are as follows.

(1) Initially, we spread the rapeseed pods throughout the image and then dispersed the coins across the far-left side. Then, after reading the image, we used Gaussian blur, grayscale processing, and threshold processing to process the image (Figure 4). Image noise was corrected based on the pixel area due to the presence of background impurities.

(2) We extracted the contour of the coin, calculated its bounding rectangle, and constructed the perimeter contour after arranging the contour points from the top left to the top right to the bottom left to the top left. The midpoint between the upper left and upper right corners was then determined by a midpoint function, which was followed by a midpoint between the bottom left and lower right corners. The midpoints of the upper left and lower left corners and the upper right and lower right corners were also calculated.

(3) Following the initialization of the measurement index value, we computed the width of the reference object in the image using Euclidean distance and exchanged the pixel points representing the area and diameter of the coin. The contour of the rapeseed pods was tested using the DL model.

(4) The length, width, and pixel points of the area were calculated using the following definitions. The length is equal to half of the irregular polygonal shape created by the rapeseed pod body and rafter; the width is determined by dividing the length of the fruit by the area of the rapeseed pod; and the area is the space encircled by the irregular polygon. Finally, we determined the value of exchanging length, width, and coin diameter pixels and rapeseed pod area pixels and coin area pixels. The formula is as follows:$$\begin{array}{c} \text{diameter\_ratio}=\text{coin\_diameter\_true}/\text{coin\_diameter\_pixel}\\ \text{area\_ratio}=\text{coin\_area\_true}/\text{coin\_area\_pixel}\\ \text{width}=\text{width\_pixel}\times \text{diameter\_ratio}\\ \text{length}=\text{length\_pixel}\times \text{diameter\_ratio}\\ \text{area}=\text{area\_pixel}\times \text{area\_ratio} \end{array}$$

The term “diameter_ratio” refers to the ratio of an RMB coin’s real diameter (coin_dimeter_true) (25 mm) to the coin’s pixel diameter (coin_dimeter_pixel) and is analogous to the term “area_ratio.” In addition, each rapeseed pod image’s width, length, and area pixels are indicated by variables, such as width_pixel, length_pixel, and area_pixel, respectively.

(5) As a final step, we depicted the estimated findings in the image (Figure 5 and Figure 6). Using regression analysis, we compared the length and width acquired by the machine vision approach with the average rapeseed pod length and width obtained after three manual measurements, respectively.

Our findings showed that the rapeseed pods ranged in length from 4.7 cm to 15.1 cm and in width from 0.2 cm to 0.5 cm when measured manually. The measured sample length had an average length of 7.7 cm, a median length of 7.4 cm, a mode length of 6.8 cm, and widths of 0.35, 0.4, and 0.37 cm. Rapeseed pod lengths, as determined by machine vision, ranged from 4.670 cm to 15.351 cm, and their widths ranged from 0.218 cm to 0.444 cm. The measured sample length had an average length of 7.7 cm, a median length of 7.3 cm, a mode length of 6.5 cm, and widths of 0.35, 0.36, and 0.37 cm. In regard to the measurement period, machine vision measurement was notably quick, whereas manual measurement consumed considerable time and labor. We calculated the correlation coefficient between manual measurement and machine vision measurement techniques using linear regression analysis. We discovered that the linear regression function was y = 0.9990x − 0.00044, and the length regression coefficient for both approaches was 0.991. y = 1.0215x − 0.0069 is the regression function, and the correlation coefficient of the width was 0.989. As a result, no substantial difference was observed in the error between manual and machine vision measurements.

Therefore, in this study, we created a novel method for rapeseed pod identification utilizing computer vision technology and DL algorithms. In addition, we created a method using machine vision to calculate the size of rapeseed pods, including their length, width, and two-dimensional image area. In comparison with the traditional hand-counting method, our proposed method has higher accuracy and precision while also being far more efficient.

## 3. Materials and Methods

### 3.1. Plant Experimental Materials and Image Acquisition

In this experiment, the “Zhejiang University 630” rapeseed variety from Zhejiang University was used and planted in the Farming Park of Zhejiang A&F University (119.72 E, 30.25 N), Lin’an District, Hangzhou City, Zhejiang Province, China. Rapeseed pod is dried in the sun after harvest. The photos were taken by a Canon EOS 800D camera with a black flocked cloth backdrop and a 70 cm lens from the table. The image resolution of the dataset is 2400 × 1600.

### 3.2. Experimental Operation Environment

A Windows 11 operating system was used in this experiment. The CPU configuration was a 12th Gen Intel(R) Core (TM) i5-12400F 2.50 GHz, and the GPU used an NVIDIA GeForce RTX 3060 graphics card with 8 GB. Anaconda3 was used to develop the training virtual environment, and the code running environment was Python 3.8, PyTorch 1.9.1, and torchvision 0.15.0. During the training process, NVIDIA CUDA11.3 was used to reduce training time. During the training phase, the image resolution was randomly adjusted to 640 × 640. We use Labelme software to manually label rapeseed pod data as a training dataset [40] (Figure 7).

### 3.3. Rapeseed Pods Data Augmentation

As we all know, a sufficient sample size will have a better effect on the training of deep learning models. Generally speaking, the larger the sample size, the better the trained model, and the stronger the generalization ability of the model. Therefore, we use image rotation and flipping, image scaling, random cropping, etc., to process rapeseed pod data. Data enhancement can increase the amount of data trained and improve the generalization ability of the model and can also increase the noise data to improve the robustness of the model.

### 3.4. YOLO v8 Model Design

YOLO v8 is a cutting-edge, state-of-the-art (SOTA) model, which builds on the success of previous YOLO versions and introduces new features and improvements to further boost performance and flexibility. As a one-stage object detection method, it has the same advantages as YOLO v1-YOLO v7. Only once does it need to extract features to achieve object detection, which is faster than other two-stage algorithms [41,42,43,44]. 

The backbone of YOLO v8 chose the C2f module instead of the C3 module, and the number of blocks per stage changed from [3, 6, 9, 3] to [3, 6, 6, 3], and its biggest change was that Anchor Base was replaced by Anchor Free, referring to the ideas of TOOD and YOLO v6 ppyoloe. YOLO v8 is extensible and supports all previous versions of YOLO’s framework; it is easy to switch between different versions and compare their performance, which makes YOLO v8 ideal for users who want to take advantage of the latest YOLO technology while still being able to use existing YOLO models. The YOLO v8 structure is shown in Figure 8. 

### 3.5. Design of a Mask R-CNN Model Based on Detectron2

Mask R-CNN, the best model of ICCV2017, is a compact and flexible universal object instance segmentation framework that can not only detect targets in the image but provide high-quality segmentation results for each target [45]. Mask R-CNN is a very flexible framework that performs well in instance segmentation tasks. Its ancestor, the R-CNN algorithm, introduced in 2014, also known as Regions with CNN Features, is a classic work that applies deep learning to object detection and greatly improves object detection performance with excellent feature extraction ability of convolutional networks. Through numerous versions of improvements, including milestone Fast R-CNN and Faster R-CNN, as well as the Mask R-CNN we use, the object detection problem has been further optimized, with progress in implementation, speed, and accuracy [46,47]. 

Mask R-CNN is an extension of Faster R-CNN and adds a new branch for predicting object masks on the bounding box recognition branch in parallel, whose structure is very similar to Faster R-CNN, but there are also three main differences: the relatively excellent Resnet FPN structure in the basic network is used because multi-layer feature maps are conducive to multi-scale object and small object detection; the RoI Align method is proposed to replace RoI Pooling to improve accuracy; and mask branch is added to predict each pixel category. The Mask R-CNN structure is shown in Figure 9. 

#### 3.5.1. Feature Extraction Network

The FPN basic network based on Resnet is a multi-layer feature combination structure, including bottom-up, top-down, and horizontal connections, which can fuse shallow, middle, and deep features, so that the features have strong semantics at the same time [48]. Mask R-CNN adds a P6 layer to the FPN, and P5 is pooled to the maximum in order to obtain a larger receptive field feature. To filter the feature map corresponding to RoI, the following formula is given:$$k=[k_{0}+\log_{2}(\sqrt{wh}/224)]$$
where k represents which feature map to take a specific anchor on, 224 represents the pre-training image size, k_0_ defaults to 4, and the corresponding level of RoI with a size of 224 × 224 is 4, which is rounded after calculation.

#### 3.5.2. RoI Align

In order to solve the miss alignment problem caused by RoI Pooling, RoI Align retains all floating points, obtains the value of multiple sample points through bilinear interpolation, and then maximizes the support of multiple sample points to obtain the final value of the point.

#### 3.5.3. Loss Task Design

Mask R-CNN adopts an FCN (Fully Convolutional Network) network structure, uses convolution and deconvolution to build an end-to-end network, and finally classifies each pixel to achieve segmentation. Also, after obtaining the characteristics of the region of interest, add the Mask branch to avoid competition between classes. The loss function is divided into three parts, and the formula is as follows:L = L_cls_ + L_box_ + L_mask_
where L_cls_ and L_box_ represent class loss and bounding box regression loss, respectively, and L_mask_ represents segmentation loss. The Sigmoid function is applied to each pixel on the mask, sent to the cross-entropy loss, and finally averaged.

Detectron2 is a software system studied by Facebook AI Research (FAIR), which inherits and reconfigures the previous Detectron and utilizes the PyTorch deep learning framework to replace the first generation of Caffe, which trains faster and integrates advanced object detection and semantic segmentation algorithms. For a large number of trained models, plug-and-play is very convenient, including end-to-end implementation of Faster R-CNN, Retina-Net, and Fast R-CNN for object detection, Mask R-CNN for instance segmentation, Key-point R-CNN for pose segmentation, and Panoptic FPN for panoramic segmentation. It contains fewer growing features than the first, such as all-optical segmentation, dense, Cascade R-CNN, a rotating bounding box, etc. [49]. 

## 4. Discussion

Pods are a crucial part of several crops. As a plant grows, the pod can shield the seed from biotic and abiotic pressures and lessen the harm caused by environmental and nonenvironmental variables. In addition, the pod may continually supply nutrients and room for the growth and development of seeds by serving as a basic photosynthetic organ during plant maturity. The observation and measurement of pod-related traits have grown in importance over the last several decades. Traditional evaluations of pod features, however, have depended on time- and labor-intensive human observations for a number of years. In this study, we created a novel method for rapeseed pod identification utilizing computer vision technology and DL algorithms. In addition, we created a method using machine vision to calculate the size of rapeseed pods, including their length, width, and two-dimensional image area. In comparison with the traditional hand-counting method, our proposed method has higher accuracy and precision and is also far more efficient.

Pod research is essential to increase seed production because seed quantity, seed weight, and the number of pods per plant are all thought to be variables influencing yield component attributes [50]. Leguminous plants and several oil crops have been the subject of extensive research on pod traits over the last three years, ranging from the use of molecular technology in breeding 3.0 to the fusion of “Biotechnology, Artificial Intelligence, and Big Data Information” technologies in breeding 4.0. Numerous novel studies focused on the use of DL and machine vision technologies to gather and analyze crop phenotypes in high throughput, which will significantly increase breeding efficiency. Li et al. suggest a DL framework to determine the phenotype of soybean maturity, which can be measured with speed, accuracy, and high throughput [51]. Through phenotypic analysis, Fabian et al. determined a specific correlation between the weight of cocoa beans and their pods [52]. Using a biotechnology technique, the researchers examined the pod color of two broad bean kinds and offered assumptions about the causes of variations in pod color brought on by chlorophyll changes [53]. Using a computed tomography scanner to capture 10 two-dimensional light projection images of peanut pods, Domhofer et al. sought to identify key factors, such as kernel weight and shell weight, that have an important influence on peanut prices [54]. The majority of pod studies are still being conducted from the perspectives of weight or color or technical utilization of commercial software. In addition, the use of DL, specifically on rapeseed pods, has not been extensively studied. Researchers had previously used ImageJ to identify comparable traits in rapeseed pods. Since this recognition method relies on sophisticated graphics software and the detection requires manual photo processing, it will be difficult to incorporate it into portable detection equipment for field trait detection in the future [55]. The technique created in this study has the ability to capture and produce pod-related phenotypic data simultaneously while processing a large number of pictures at high throughput. This can greatly improve research efficiency.

In this paper, we developed an instance segmentation approach for rapeseed pods in *Brassica napus* and utilized for the first time the popular DL Mask R-CNN model based on Detectron2 and YOLO v8, which was not released until January 10, 2023. The YOLO series of models, such as the variant of YOLO v5, has made progress in road defect detection. Machine learning, like Graph Cooperative Learning Neural Networks, can perform better on data augmentation [56,57]. According to our findings, YOLO v8 and Mask R-CNN trained extremely effectively, with YOLO v8 showing a marginal outperformance. Mask R-CNN exhibited higher segmentation and recognition rates, which reached nearly 100%, on the test dataset of freshly harvested rapeseed pods from various types or growth stages. In this study, Mask R-CNN was developed on Detectron2, a library that combines a number of pre-training deep learning models, including Faster R-CNN, Retina Net, and Panoptic FPN. In this experiment, the Detectron2-based system surpasses YOLO v8 with its tiny and quick characteristics in terms of speed and accuracy. Detectron2 is typically used to deliver Excellent accuracy and a speedy training speed. The application of Detectron2 made rapeseed pod phenotypic data more useful for use in field research by minimizing the complexity of space and significantly reducing the computing time. Regarding AI security and maintenance costs, Detectron2 currently presents a more favorable outlook due to its swifter iteration pace and more frequent updates. The adoption of an estimator instead of Slim has led to increased code complexity, but it has also resulted in nearly comprehensive test coverage and standardized annotations [58,59,60,61]. Furthermore, because Detectron2 incorporates several widely employed deep learning models for object detection and instance segmentation, it possesses the potential for future compatibility with a broader range of agricultural and industrial production scenarios. These scenarios may include tasks like recognizing plant fructifications and identifying crop pests, extending its applicability beyond the sole measurement of rapeseed pod phenotype omics data [62,63,64,65,66]. By combining machine vision, we also determined the length, width, and two-dimensional image area of the rapeseed pods in the image using a single coin as a reference. The advantages of our approach over the conventional manual measurement method include minimal labor costs, quick calculation times, and high accuracy.

## 5. Conclusions and Future Work

In conclusion, we have successfully developed an innovative DL-based approach for the segmentation and collection of phenotypic data related to rapeseed pods. This represents a valuable addition to the existing methodologies employed in rapeseed pod analysis. Both of our models have demonstrated strong performance in recognizing rapeseed pods. Nevertheless, it is important to acknowledge the limitations of our method. For instance, our research has indicated that accurately calculating the length, width, and area of rapeseed pods with significant curvature remains a challenge. Our strategy is more effective when applied to rapeseed pods with less pronounced curvature. Additionally, our research necessitates the physical harvesting of rapeseed pods for identification, and we aspire to explore the possibility of directly identifying and quantifying phenotypic traits of rapeseed pods in field conditions in the future.

As modern agricultural practices increasingly embrace mechanization, our research findings hold the potential to significantly benefit the field of rapeseed phenotype recognition. This could lead to the optimization of mechanized rapeseed harvesting processes, enhancing the efficiency of rapeseed industrial agricultural product production, including protein and oil. Furthermore, by expanding our dataset, testing a wider range of rapeseed varieties, and improving the environmental conditions for rapeseed growth during testing, we may extend the applicability of our approach to future genomics analyses of rapeseed pod characteristics. This methodology could also prove valuable for genome-wide association studies of rapeseed pod properties in planting fields.

Our approach offers breeders an effective means of digitally analyzing phenotypes and automating the identification and screening processes, not only for rapeseed germplasm resources but also for pod-bearing leguminous plants, like soybeans.

## Figures and Tables

**Figure 1 plants-12-03328-f001:**
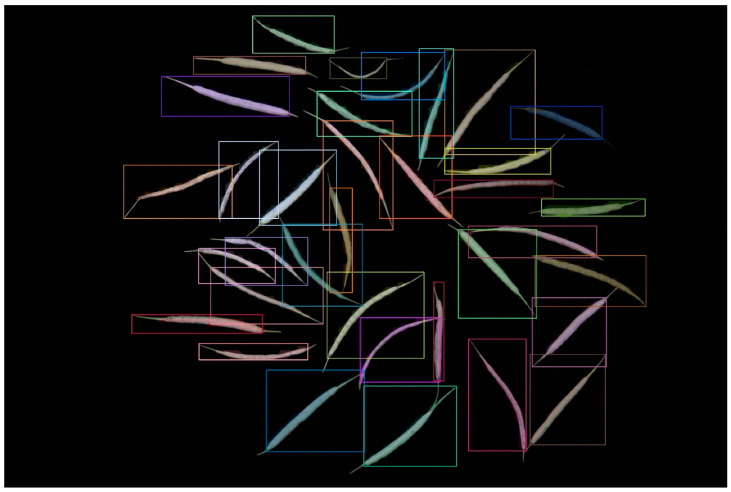
Result of the YOLOv8 instance segmentation.

**Figure 2 plants-12-03328-f002:**
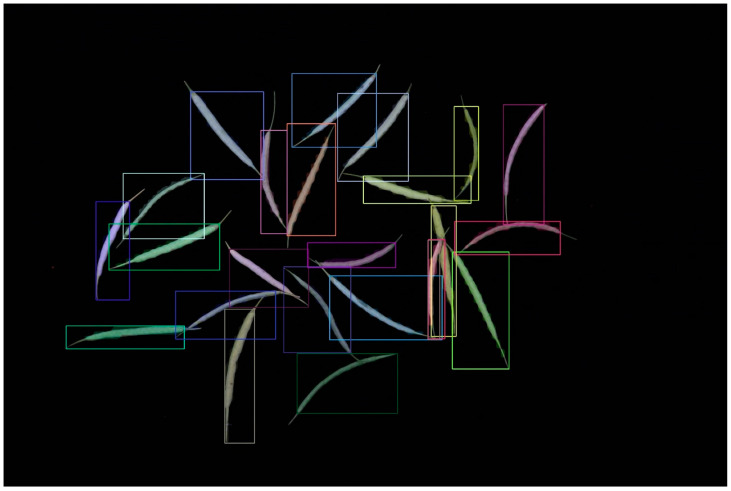
Instance segmentation result of using Mask-R-CNN based on Detectron2.

**Figure 3 plants-12-03328-f003:**
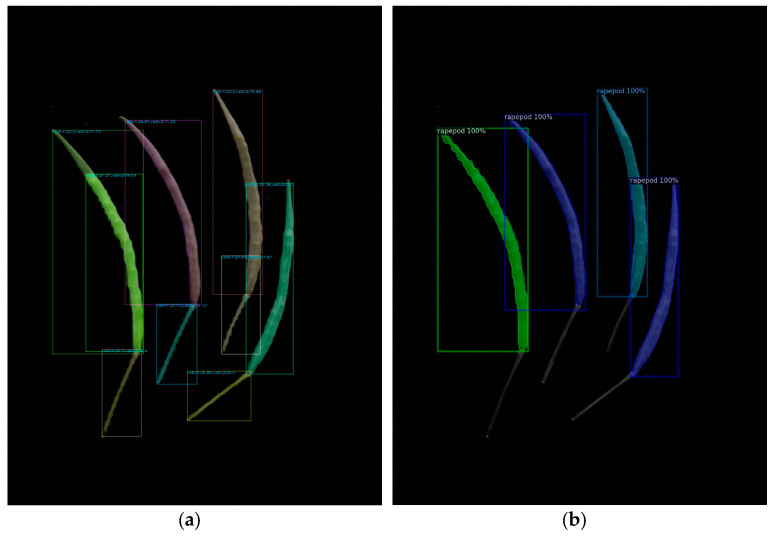
Test result of the deep learning models: (**a**) YOLO v8; (**b**) Mask R-CNN.

**Figure 4 plants-12-03328-f004:**
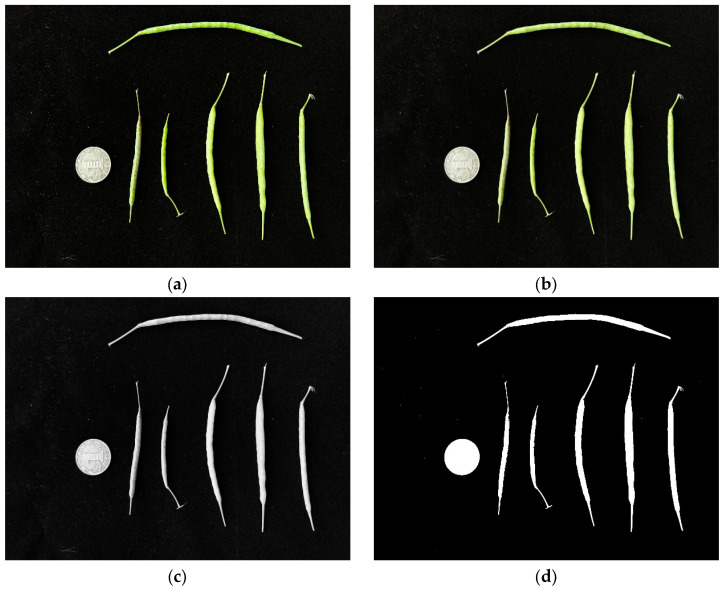
Preprocessing of rapeseed pod measurement images. (**a**) Raw, (**b**) Gaussian blur, (**c**) grayscale processing, and (**d**) threshold processing images.

**Figure 5 plants-12-03328-f005:**
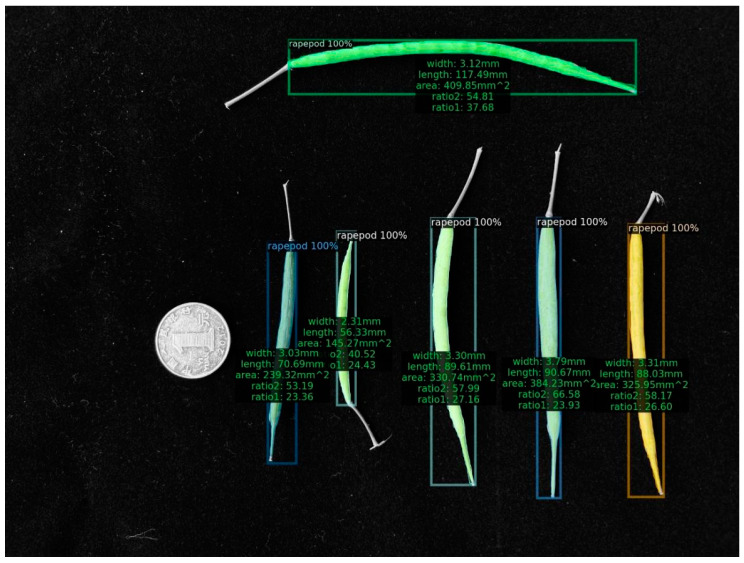
Image instance of the computed results.

**Figure 6 plants-12-03328-f006:**
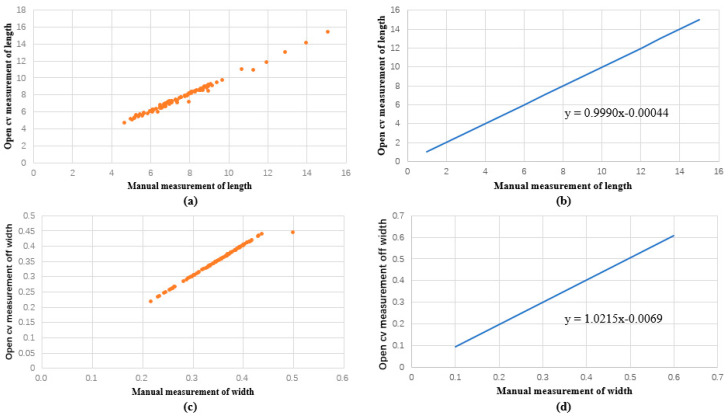
Comparison between manual and machine vision of the length and width of the rapeseed pods obtained in regression function images. (**a**) Discrete relationship between open cv and manual measurement of length. (**b**) Linear function image of the relationship between open cv measurement and manual measurement of length. (**c**) Discrete relationship between open cv and manual measurement of width. (**d**) Linear function image of the relationship between open cv measurement and manual measurement of width.

**Figure 7 plants-12-03328-f007:**
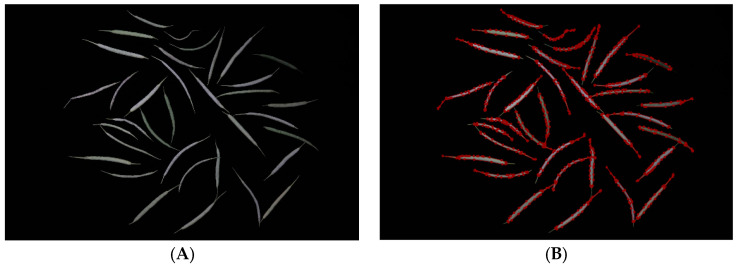
(**A**) A raw image of a rapeseed pod training set; (**B**) an image of a rapeseed pod training set annotated by Labelme software.

**Figure 8 plants-12-03328-f008:**
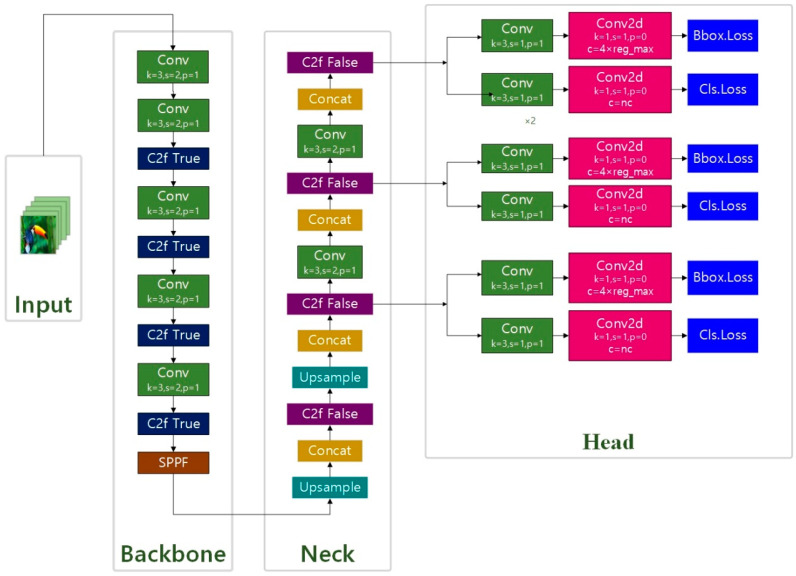
The model structure of YOLO v8.

**Figure 9 plants-12-03328-f009:**
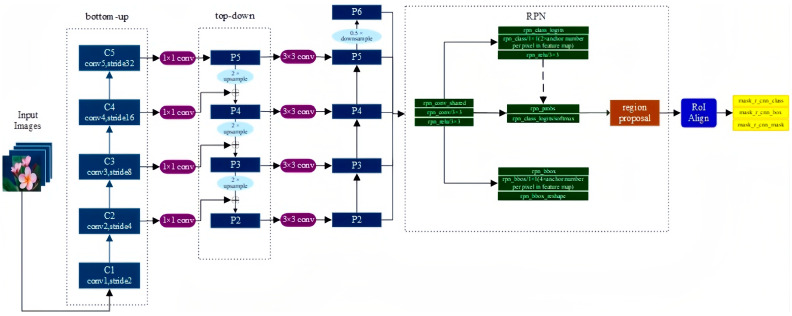
The structure of Mask R-CNN.

**Table 1 plants-12-03328-t001:** YOLOv8 model evaluation metrics.

Model	Box				Mask				FLOPs (B)	Params (M)	Gradients
	P	R	mAP50	mAP50-95	P	R	mAP50	mAP50-95			
YOLOv8n	0.985	0.991	0.991	0.927	0.985	0.987	0.991	0.742	12.0 G	3.41 M	3,409,952
YOLOv8s	0.992	0.998	0.991	0.963	0.99	0.996	0.991	0.782	42.7 G	11.79 M	11,790,467
YOLOv8m	0.995	1.000	0.991	0.972	0.993	0.998	0.991	0.790	110.4 G	27.24 M	27,240,211

**Table 2 plants-12-03328-t002:** Model evaluation metrics under different training iterations and backbones.

Backbone		ITER	AP	AP50	AP75	Aps	APm	Apl
Resnet50	bbox	10,000	89.788	97.923	96.916	Nan	91.980	85.866
	segm		74.620	97.900	95.828	Nan	74.559	76.652
	bbox	30,000	91.409	97.829	96.711	Nan	93.267	86.174
	segm		75.232	97.856	96.804	Nan	75.216	77.161
Resnet101	bbox	10,000	91.132	97.924	96.650	Nan	93.168	86.364
	segm		75.129	97.909	95.712	Nan	75.113	76.596
	bbox	30,000	92.481	97.997	96.680	Nan	94.356	89.223
	segm		75.465	97.774	95.603	Nan	75.134	78.263

**Table 3 plants-12-03328-t003:** Precision and test time spent by the DL model.

Model	YOLO v8	Mask R-CNN
Precision	91.263	99.181
Test time (Unit: s)	108.97	46.30

## Data Availability

The data presented in this study are available upon request from the corresponding author. The data used in our research are not publicly available, as they are also being utilized in an ongoing study.

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
