# Peer review of "Segmentation and Phenotype Calculation of Rapeseed Pods Based on YOLO v8 and Mask R-Convolution Neural Networks"

_plants, 2023, doi:10.3390/plants12183328_

Round 1
Reviewer 1 Report
Dear Authors,
the article is quite interesting and relevant, however, its structure is strange, and some notes should be revised before promotion:
1. Current section "4. Materials and Methods" should be after section 1 "Introduction", while section "3. Discussion" looks like summary of the methods analysis.
2. Section "2. Results" should finish the presentation of the study; after that the new Discussion section should be placed. Real Discussion part should contain clear stated Authors' advances over existing studies, limitations of research and its future perspectives.
3. In the Section "1 Introduction", please introduce the concept of your method of research of neural networks for analysis of rapeseed pods cultivation, emphasizing the significance for energy industry development in China.
4. In Materials and Methods section, please explain the choice of YOLO v8 and Mask R CNNs from the variety of neural networks. Please give clear view on the methodology used in the research model, selection criteria for relevant works, and data analysis methods.
5. In Results section, please give clear description of case conditions and specific.
Good luck!
Review the manuscript for grammatical errors and ensure consistency in terminology and expression throughout the text.
Consider rephrasing sentences for improved coherence and flow of ideas.
Author Response
From the Reviewer #1:
Comments and Suggestions for Authors:
the article is quite interesting and relevant, however, its structure is strange, and some notes should be revised before promotion:
Response: We are appreciated for the positive comments. We have modified the entire manuscript according to your valuable suggestions. Please see the following:
Q1. The current section "4. Materials and Methods" should be after section 1 "Introduction", while section "3. Discussion" looks like a summary of the methods analysis.
Response: Thank you for your valuable suggestions. Initially, we structured the manuscript according to the magazine's template. In this revised version, we have incorporated your recommendations to adjust the article's structure. Additionally, we have extensively revised the discussion section and made numerous additions to enhance its content.
Q2. Section "2. Results" should finish the presentation of the study; after that, the new Discussion section should be placed. The real Discussion part should contain clearly stated Authors' advances over existing studies, limitations of research, and its future perspectives.
Response: We greatly appreciate your valuable suggestions. As per your recommendations, we have repositioned the initial paragraph of the discussion section to the conclusion of the results section, thereby providing a more coherent presentation of our study. In the discussion and conclusion section, we have now clearly outlined the advancements made compared to existing studies, acknowledged the research's limitations, and articulated our future perspectives. (Line356-361).
Q3. In the Section "1 Introduction", please introduce the concept of your method of research of neural networks for analysis of rapeseed pods cultivation, emphasizing the significance for energy industry development in China.
Response: Thanks for your good suggestions. Based on your suggestion, we have made the following modifications and text additions in the introduction part to introduce the concept of our method and emphasize the significance of energy industry development in China.
First part: Line41-49: Rapeseed oil is also a valuable feedstock for biofuel production, particularly biodiesel. The conversion of rapeseed oil into biodiesel offers a cleaner and more sustainable alternative to fossil fuels, contributing to the reduction of greenhouse gas emissions and air pollution. China, being one of the world's largest energy consumers, has a vested interest in expanding its biofuel production capacity to mitigate environmental challenges and reduce its reliance on imported fossil fuels. However, with the growing demand for renewable energy sources and sustainable agricultural practices, the cultivation of rapeseed pods has gained renewed attention in the context of the energy industry in China.
Second part: Line50-52 “The value of rapeseed seeds, which are a raw material in the oil industry, is strictly dependent on varietal yield parameters, which are one of the most important elements influencing crop production in China[10-12]”. we emphasized the significance of energy industry development in China.
Third part: To highlight the significance of our study, we have incorporated some recent findings on the use of neural networks for rapeseed.
“In addition, Du et al. presented the plant segmentation transformer network to segment dense point cloud data with complicated spatial structures gathered by handheld laser scanning in the pod stage of rapeseed[23]. Han et al. invented a deep learning net called InceptionV3-LSTM for the intelligent prediction of rapeseed harvest time and a convolutional neural network model was applied for variety classification and seed quality assessment of winter rapeseed[24,25].” (Line73-78)
Fourth part: Line96-99: At present, there are still the following problems in the cultivation of rapeseed pods, such as the lack of effective monitoring methods for the quantity and quality of rapeseed pods, and a lack of in-depth understanding of the growth process and influencing factors of rapeseed pods.
Fifth part: Line112-114: This study applied neural networks to analyze the phenotypic data information of rapeseed pods, which can be useful for future industrial production.
Q4. In Materials and Methods section, please explain the choice of YOLO v8 and Mask R-CNNs from the variety of neural networks. Please give clear view on the methodology used in the research model, selection criteria for relevant works, and data analysis methods.
Response: Thanks for your good suggestions. The reason we choose YOLO v8 and Mask R-CNN is that YOLO v8 is the latest YOLO series model which was published in 2023, And Mask R-CNN is the most effective instance segmentation model. We added the reason in the materials and methods part. (Line143-144, Line159-162)
Q5. In Results section, please give clear description of case conditions and specific.
Response: Thanks for your good suggestions. Based on your suggestions, we have made the following illustration and text addition in the results section:
Line215- 219: We collected 4461 rapeseed pods and randomly placed them on a black velvet background. We used a Canon EOS 800D camera to capture RGB images and enhance the data of the images, such as flipping, rotating, cropping, adding noise, blurring, masking, color transformation, cutout, and other methods, which aim was to increase the model's generalization ability and avoid sample imbalance.
Line236-237: We utilized the identical dataset with YOLO v8 and performed tasks akin to those in section 3.1 of our image processing work.
Comments on the Quality of English Language: Review the manuscript for grammatical errors and ensure consistency in terminology and expression throughout the text.
Consider rephrasing sentences for improved coherence and flow of ideas.
Response:
We are very appreciative of your suggestions, particularly the efforts of the detail modification tips which are very useful for helping us to correct errors and improve the quality of our manuscript. We apologize for our grammatical errors and incoherent sentences. Meanwhile, we have carefully corrected the grammar errors in the sentence to make it more coherent and authentic.
Reviewer 2 Report
This paper deals with an exciting topic. The article has been read carefully, and some minor issues have been highlighted in order to be considered by the author(s).
#1 What is the motivation of this paper?
#2 What is the contribution and novelty of this paper?
#3 What is the advantage of this paper?
#4 Which evaluation metrics did you used for comparison?
#5 It would be good if AI model security issues would be reflected in the related work such as “Adversarial image perturbations with distortions weighted by color on deep neural networks”, “Dual-Mode Method for Generating Adversarial Examples to Attack Deep Neural Networks”, "Toward Selective Adversarial Attack for Gait Recognition Systems Based on Deep Neural Network”, “Priority Evasion Attack: An Adversarial Example That Considers the Priority of Attack on Each Classifier”, “Friend-guard adversarial noise designed for electroencephalogram-based brain–computer interface spellers”.
#6 Author can clearly explain the challenges faced in the existing system and the motivation of the proposed system.
#7 Meaning of the symbols used can be explained clearly.
#8 The limitation of the proposed work can be discussed.
Author Response
From the reviewer #2
Comments and Suggestions for Authors:
This paper deals with an exciting topic. The article has been read carefully, and some minor issues have been highlighted in order to be considered by the author(s).
Response:
We are very appreciative of your professional review work on our article. We sincerely appreciate the positive comments. As you are concerned, all the amendments have been addressed. According to your suggestions, we have made extensive corrections to our previous draft, the detailed corrections are listed below.
#1 What is the motivation of this paper?
Response: Thank you for your good question. The motivation for this paper is as follows:
- Rapeseed is an important oil plant all around the world (Line38-41)
- At present, there are still the following problems in the cultivation of rapeseed pods, such as the lack of effective monitoring methods for the quantity and quality of rapeseed pods, and a lack of in-depth understanding of the growth process and influencing factors of rapeseed pods. (Line96-99)
- This study applied neural networks to analyze the phenotypic data information of rapeseed pods, which can be useful for future industrial production. Meanwhile, the study methodologies described in this publication can provide a reference and guidance for research on other crops. (Line112-115)
#2 What is the contribution and novelty of this paper?
Response: Thanks for your good questions. We added the novelty in several parts of this manuscript.
For the novelty, please see Line103-108: In this paper, we propose a simple and efficient method for calculating the number, length, width, and two-dimensional image area of rapeseed pods by DL and computer vision and observing the length patterns of rapeseed pods during different time periods, such as the green ripening, yellow ripening, and mature periods. It was the first time that the most recent deep learning algorithms, YOLO v8 and Mask R-CNN, had been employed to detect yield correlation in rapeseed.
For the contribution, please see Line26-29: Our approach offers breeders an effective strategy for digitally analyzing phenotypes, automating the identification and screening process, not only in rapeseed germplasm resources but also in leguminous plants like soybeans that possess pods.
#3 What is the advantage of this paper?
Response: Thanks for your good questions. I am sorry for not highlighting the parts you mentioned in our previous article. We have made modifications according to your suggestions. The advantage of this paper is as follow:
Line356-361: In this study, we created a novel method for rapeseed pod identification utilizing computer vision technology and DL algorithms. In addition, we created a method using machine vision to calculate the size of rapeseed pods, including their length, width, and two-dimensional image area. In comparison with the traditional hand-counting method, our proposed method has higher accuracy and precision while also being far more efficient.
Line397-403: Researchers had previously used ImageJ to identify comparable traits in rapeseed pods. Since this recognition method relies on sophisticated graphics software and the detection requires manual photo processing, it will be difficult to incorporate it into portable detection equipment for field trait detection in the future[54]. The technique created in this study has the ability to capture and produce pod-related phenotypic data simultaneously while processing a large number of pictures at high throughput. This can greatly improve research efficiency.
Line404-406: In this paper, we developed an instance segmentation approach for rapeseed pods in Brassica napus and utilized for the first time the popular DL Mask R-CNN model based on Detectron2 and YOLO v8, which was not released until January 10, 2023.
Line413-419: In this study, Mask R-CNN was developed on Detectron2, a library that combines a number of pre-training deep learning models, including Faster R-CNN, Retina Net, and Panoptic FPN. In this experiment, the Detectron2-based system surpasses YOLO v8 with its tiny, quick characteristics in terms of speed and accuracy. Detectron2 is typically used to deliver Excellent accuracy and a speedy training speed. The application of Detectron2 made rapeseed pod phenotypic data more useful for use in next field research by minimizing the complexity of space and significantly reducing the computing time.
Line429-432: By combining machine vision, we also determined the length, width, and two-dimensional image area of the rapeseed pods in the image using a single coin as a reference. The advantages of our approach over the conventional manual measurement method include minimal labor costs, quick calculation times, and high accuracy.
#4 Which evaluation metrics did you used for comparison?
Response: Thanks for the suggestions. We mainly used precision confidence (P) and test time to compare different models. Please refer to Table 1, Table 2, and Table 3 for detailed information.
#5 It would be good if AI model security issues would be reflected in the related work such as “Adversarial image perturbations with distortions weighted by color on deep neural networks”, “Dual-Mode Method for Generating Adversarial Examples to Attack Deep Neural Networks”, "Toward Selective Adversarial Attack for Gait Recognition Systems Based on Deep Neural Network”, “Priority Evasion Attack: An Adversarial Example That Considers the Priority of Attack on Each Classifier”, “Friend-guard adversarial noise designed for electroencephalogram-based brain–computer interface spellers”.
Response: Thanks for the valuable suggestions and good recommendation of the related papers. According to your suggestion, we have made the following modifications and the suggested papers were properly cited:
Line419-423: Regarding AI security and maintenance costs, Detectron2 currently presents a more favorable outlook due to its swifter iteration pace and more frequent updates. The adoption of an estimator instead of Slim has led to increased code complexity, but it has also resulted in nearly comprehensive test coverage and standardized annotations[59-62].
#6 Author can clearly explain the challenges faced in the existing system and the motivation of the proposed system.
Response: Thanks for the suggestions. According to your suggestion, we have made the following corrections and explanations:
Line438-441: Nevertheless, it is important to acknowledge the limitations of our method. For instance, our research has indicated that accurately calculating the length, width, and area of rapeseed pods with significant curvature remains a challenge. Our strategy is more effective when applied to rapeseed pods with less pronounced curvature.
Line445-453: As modern agricultural practices increasingly embrace mechanization, our research findings hold the potential to significantly benefit the field of rapeseed phenotype recognition. This could lead to the optimization of mechanized rapeseed harvesting processes, enhancing the efficiency of rapeseed industrial agricultural product production, including protein and oil. Furthermore, by expanding our dataset, testing a wider range of rapeseed varieties, and improving the environmental conditions for rapeseed growth during testing, we may extend the applicability of our approach to future genomics analyses of rapeseed pod characteristics. This methodology could also prove valuable for genome-wide association studies of rapeseed pod properties in planting fields.
#7 Meaning of the symbols used can be explained clearly.
Response: Thanks for your good suggestions. We have made the modification according to your suggestions.
#8 The limitation of the proposed work can be discussed.
Response: Thanks for the suggestions. We discussed the limitations of the work in the paper:
Line438-444: Nevertheless, it is important to acknowledge the limitations of our method. For instance, our research has indicated that accurately calculating the length, width, and area of rapeseed pods with significant curvature remains a challenge. Our strategy is more effective when applied to rapeseed pods with less pronounced curvature. Additionally, our research necessitates the physical harvesting of rapeseed pods for identification, and we aspire to explore the possibility of directly identifying and quantifying phenotypic traits of rapeseed pods in field conditions in the future.
Reviewer 3 Report
The authors present Segmentation and phenotype calculation of rapeseed pods based on YOLO v8 and Mask R-CNN . The study is interesting. In general, the main conclusions presented in the paper are supported by the figures and supporting text. However, to meet the journal quality standards, the following comments need to be addressed.
• Abstract: Should be improved and extended. The authors talk lot about the problem formulation, but novelty of the proposed model is missing. Also provided the general applicability of their model. Please be specific what are the main quantitative results to attract general audiences.
• The introduction can be improved. The authors should focus on extending the novelty of the current study. Emphasize should be given in improvement of the model (in quantitative sense) compared to existing state-of-the art models.
• More details about network architecture and complexity of the model should be provided.
• what about comparison of the result with current state-of-the art models? Did authors perform ablation study to compare with different models?
• What are the baseline models and benchmark results? The authors may compared the result with existing models evaluated with datasets
• Conclusion parts needs to be strengthened.
• Please provide a fair weakness and limitation of the model, and how it can be improved.
• Typographical errors: There are several minor grammatical errors and incorrect sentence structures. Please run this through a spell checker.
Discussions of relevant literature could be further enhanced, which can help better motivate the current study and link to the existing work. Authors might consider the following relevant recent work in the field of applying computer vision techniques to better motivate the usefulness of machine learning approaches, such as
see : -
object detection
- Neural Networks 2022 https://doi.org/10.1016/j.neunet.2022.05.024
YOLO
-Adv. Eng. Informatics 2023, 56, 102007, https://doi.org/10.1016/j.aei.2023.102007
Hence they should be briefly discussed in the related work section.
See above
Author Response
From the reviewer #3
Comments and Suggestions for Authors
The authors present Segmentation and phenotype calculation of rapeseed pods based on YOLO v8 and Mask R-CNN. The study is interesting. In general, the main conclusions presented in the paper are supported by the figures and supporting text. However, to meet the journal quality standards, the following comments need to be addressed.
- Abstract: Should be improved and extended. The authors talk a lot about the problem formulation, but the novelty of the proposed model is missing. Also provided the general applicability of their model. Please be specific what are the main quantitative results to attract general audiences.
Response: We appreciate your positive comments and practical suggestions. Based on your suggestions, we have made the following modifications and text additions to the abstract.
Line16-18: The YOLO v8n model and the Mask R-CNN model with a Resnet101 backbone in Detectron2 both achieve precision rates exceeding 90%.
Line24-29: In conclusion, for the first time we utilized deep learning techniques to identify the characteristics of rapeseed pods while concurrently establishing a dataset for rapeseed pods, our suggested approaches were successful in segmenting and counting rapeseed pods precisely. Our approach offers breeders an effective strategy for digitally analyzing phenotypes, automating the identification and screening process, not only in rapeseed germplasm resources but also in leguminous plants like soybeans that possess pods
- The introduction can be improved. The authors should focus on extending the novelty of the current study. Emphasize should be given in improvement of the model (in quantitative sense) compared to existing state-of-the art models.
Response: Thanks for the good suggestions. According to your suggestions, we have made the following modifications to improve the introduction part.
Line103-108: In this paper, we propose a simple and efficient method for calculating the number, length, width, and two-dimensional image area of rapeseed pods by DL and computer vision and observing the length patterns of rapeseed pods during different time periods, such as the green ripening, yellow ripening, and mature periods. It was the first time that the most recent deep learning algorithms, YOLO v8 and Mask R-CNN, had been employed to detect yield correlation in rapeseed.
- More details about network architecture and complexity of the model should be provided.
Response: Thanks for the good question. Our research focuses on applying deep learning models to calculate the different phenotypes of rapeseed pods. After using YOLO v8 and Mask R-CNN, we found that its accuracy has exceeded 90%, achieving our expected goal. Therefore, we did not further optimize the model. However, in the early stage, we performed data enhancement preprocessing on the image data of the training model, such as image rotation and flipping, image scaling, and random cropping, and conducted Gaussian blur, grayscale processing, and threshold processing to process the image during specific phenotype calculations.
- what about comparison of the result with current state-of-the art models? Did authors perform ablation study to compare with different models?
Response: Thanks for the good suggestions. We used the latest model YOLO v8 of the YOLO series, to calculate the length, width, and area of rapeseed pods. It is the first time deep learning models have been used in yield correlation detection in rapeseed. We compared different YOLO v8 models like YOLO v8s, YOLO v8n, YOLO v8m, and Mask R-CNN based on Detectron2 including Resnet50 and Resnet101 backbone on different iterations.
- What are the baseline models and benchmark results? The authors may compared the result with existing models evaluated with datasets
Response: Thank you for your thoughtful inquiries. We have already adopted the most recent YOLO model, YOLO v8, as it has evolved from earlier YOLO iterations such as YOLO v5, which was created several years prior. Therefore, we anticipate that utilizing the YOLO v5 version would yield comparable outcomes.
- Conclusion parts needs to be strengthened.
Response: We are appreciated for your good suggestion and we have extensively modified the discussion and conclusion part:
Line433-456:
- Conclusion and future work
In conclusion, we have successfully developed an innovative DL-based approach for the segmentation and collection of phenotypic data related to rapeseed pods. This represents a valuable addition to the existing methodologies employed in rapeseed pod analysis. Both of our models have demonstrated strong performance in recognizing rapeseed pods. Nevertheless, it is important to acknowledge the limitations of our method. For instance, our research has indicated that accurately calculating the length, width, and area of rapeseed pods with significant curvature remains a challenge. Our strategy is more effective when applied to rapeseed pods with less pronounced curvature. Additionally, our research necessitates the physical harvesting of rapeseed pods for identification, and we aspire to explore the possibility of directly identifying and quantifying phenotypic traits of rapeseed pods in field conditions in the future.
As modern agricultural practices increasingly embrace mechanization, our research findings hold the potential to significantly benefit the field of rapeseed phenotype recognition. This could lead to the optimization of mechanized rapeseed harvesting processes, enhancing the efficiency of rapeseed industrial agricultural product production, including protein and oil. Furthermore, by expanding our dataset, testing a wider range of rapeseed varieties, and improving the environmental conditions for rapeseed growth during testing, we may extend the applicability of our approach to future genomics analyses of rapeseed pod characteristics. This methodology could also prove valuable for genome-wide association studies of rapeseed pod properties in planting fields.
Our approach offers breeders an effective means of digitally analyzing phenotypes, automating the identification and screening processes, not only for rapeseed germplasm resources but also for pod-bearing leguminous plants like soybeans.
- Please provide a fair weakness and limitation of the model, and how it can be improved.
Response: Thanks for the good suggestions. We have added the related information in the discussion and conclusion part.
Line438-444: Nevertheless, it is important to acknowledge the limitations of our method. For instance, our research has indicated that accurately calculating the length, width, and area of rapeseed pods with significant curvature remains a challenge. Our strategy is more effective when applied to rapeseed pods with less pronounced curvature. Additionally, our research necessitates the physical harvesting of rapeseed pods for identification, and we aspire to explore the possibility of directly identifying and quantifying phenotypic traits of rapeseed pods in field conditions in the future.
- Typographical errors: There are several minor grammatical errors and incorrect sentence structures. Please run this through a spell checker.
Response: We are very appreciated for your suggestions, particularly the efforts of the detail modification tips which are very useful for helping us to correct errors and improve the quality of our manuscript. We apologize for our grammatical errors and incoherent sentence structures. Meanwhile, we have carefully corrected the grammar errors in the sentence to make it more coherent and authentic.
Discussions of relevant literature could be further enhanced, which can help better motivate the current study and link to the existing work. Authors might consider the following relevant recent work in the field of applying computer vision techniques to better motivate the usefulness of machine learning approaches, such as
see : -
object detection
- Neural Networks 2022 https://doi.org/10.1016/j.neunet.2022.05.024
YOLO
-Adv. Eng. Informatics 2023, 56, 102007, https://doi.org/10.1016/j.aei.2023.102007
Hence they should be briefly discussed in the related work section.
Response: Thanks for the valuable suggestions. We have made the following modifications in the discussion part.
Line406-409: The YOLO series of models, such as the variant of YOLO v5, have made progress in road defect detection. Machine learning, like Graph Cooperative Learning Neural Networks, can perform better on data augmentation[57,58].
Round 2
Reviewer 1 Report
Good luck!
Reviewer 2 Report
I recommend the acceptance.
Reviewer 3 Report
The revised manuscript is now suitable for publication